# Acute Effect of Three Aerobic Exercise Intensities on Glomerular Filtration Rate in Healthy Older Adults

**DOI:** 10.3390/diseases12100249

**Published:** 2024-10-11

**Authors:** Marina Trejo-Trejo, Luis M. Gómez-Miranda, Arnulfo Ramos-Jiménez

**Affiliations:** 1Sports Faculty, Autonomous University of Baja California, Mexicali 21289, Baja California, Mexico; marina.trejo@uabc.edu.mx (M.T.-T.); luis.gomez@uabc.edu.mx (L.M.G.-M.); 2Department of Health Sciences, Biomedical Sciences Institute, Autonomous University of Ciudad Juarez, Chihuahua 32310, Chihuahua, Mexico

**Keywords:** aerobic exercise, cardiovascular risk factors, chronic kidney dysfunction, glomerular filtration rate, sports medicine

## Abstract

No consensus exists on whether acute aerobic exercise alters the glomerular filtration rate in older adults. Objective: To assess the immediate effects of three aerobic exercise intensities on the estimated glomerular filtration rate (eGFR) in healthy, sedentary older adults. Methods: Eighteen healthy, sedentary older adults (ten men and eight women) voluntarily participated in this study. The participants underwent three standardized aerobic exercise tests (100%, 80%, and 60% of the maximal heart rate) on a bicycle ergometer. Blood samples were collected to determine cholesterol, triacylglycerols, glucose, serum creatinine (Cr), Cystatin C (CysC) concentrations, and eGFR. Results: eGFR and serum concentrations of Cr and CysC were not modified at any exercise intensity. There was a negative correlation between blood total cholesterol vs. eGFR (R = −0.512, R = −0.582, R = −0.531; *p* < 0.05) at rest, 60%, and 100% of the maximal heart rate, respectively. In addition, a negative correlation existed for age vs. eGFR at 60% of the maximal heart rate (R = −0.516; *p* < 0.05). Conclusions: Short-duration aerobic exercise of low, moderate, and vigorous intensity did not significantly affect eGFR and is considered safe for kidney function in healthy, sedentary older adults. However, regular monitoring of kidney function in older people engaged in moderate- and high-intensity exercise is advised.

## 1. Introduction

Chronic kidney disease (CKD) is a global health problem with a high prevalence worldwide (~12%) [1], with 21.4% in older adults [2]. Low- and moderate-intensity chronic aerobic exercise has been proven to benefit these patients, decreasing complications and improving kidney function [3]. In addition, Forsse et al. (2023) reported temporary improvements in renal filtration following short-term (30 min) moderate- and high-intensity exercise in adults with moderate-stage (stages G3a–b) CKD [4]. On the contrary, some studies have reported that a single high-intensity resistance training session causes acute kidney damage in healthy young volunteers [5], and others observe a temporary decrease in glomerular filtration rate (GFR) in sedentary, active adults over 64 years [6]. These two latest studies indicate the possibility that high-intensity acute exercise may promote kidney damage in adults with CKD, exacerbated when exercise is performed for a long time and in dehydrating climatic conditions [7,8]. As can be noticed, there is no consensus on the effect of high-intensity acute exercise on glomerular function. Further, the immediate effects of three different exercise intensities (high, moderate, and low) on the renal function of older adults still need to be studied.

Walk-around conventional methods for assessing kidney function in older adults often need to be revised. The estimated glomerular filtration rate (eGFR) is currently the recommended technique for earlier detection and better management of CKD in this population because GFR indicates the natural kidney dysfunction associated with aging [9].

There are several methods available for calculating GFR in humans: the 3 h volume of distribution method, iohexol measured by HPLC-UV [10], the simplified one-compartment model corrected by the Bröchner-Mortensen formula [11], the camera-based method using 99mTc-DTPA [12], and p-amino hippuric acid clearance [13]. The most suitable methods for estimating GFR in clinical practice are serum creatinine (Cr) and cystatin C (CysC) concentrations. According to Laterza et al. (2002), CysC is considered a more reliable and accurate marker than Cr [14]; both methods have been utilized to assess kidney damage from acute intense exercise, such as after an ultramarathon [15].

Hence, this study aimed to assess the immediate effects of three aerobic exercise intensities on eGFR in healthy, sedentary older adults using serum concentrations of Cr and CysC. Understanding GFR responses to different exercise intensities can encourage physicians and sports coaches to collaborate in designing the physical activities and exercise intensities most suitable for the patient’s clinical characteristics—especially those who may require closer monitoring due to their physical condition.

## 2. Materials and Methods

### 2.1. Participants

Sixty older adults were invited to participate in a cross-sectional study for convenience: 35 did not meet the inclusion criteria, 5 did not complete the tests, and 2 voluntarily left the study. Finally, 18 participants (10 men and 8 women) completed the study (Figure 1). The inclusion criteria were 65 years or older, non-smoker, no evidence of kidney or liver dysfunction, and engaging in less than five days of moderate-intensity physical exercise or less than 30 min of walking daily. The exclusion criteria included inability to follow protocol instructions, mobility issues, and failure to perform a maximum exercise test on a bicycle ergometer.

### 2.2. Ethical Considerations

The Autonomous University of Baja California Ethics Committee approved this study (UABC-998/2020-2), which was conducted following the World Medical Association’s Declaration of Helsinki guidelines [16] and supervised by a sports physician. Before the study, participants voluntarily signed a written informed consent form.

### 2.3. Research Protocol

After an overnight fast, participants were required to visit the laboratory four times between 7:00 and 9:00 h (Figure 2). In the first session, a venous blood sample was collected, and a clinical history examination, a resting electrocardiogram, and a body fat percentage measurement (%BF) were performed. The second session involved a maximum exercise test under a standardized protocol [17]; the third and fourth sessions included two submaximal exercise tests at 80% or 60% of the maximal heart rate achieved during the maximum exercise test. Before each exercise test, participants drank 150 mL of water. The indoor environmental conditions during the exercise tests remained stable, with a temperature of 23 °C and relative humidity of 30%. In addition, the participants were instructed to wear sports attire.

#### 2.3.1. First Session

Medical history, laboratory, and biochemical tests were carried out to determine the participant’s health status and to prevent possible damage to their health when subjected to intense exercise. Each participant underwent a thorough examination by a physician, including a twelve-lead resting electrocardiogram (Welch Allyn 50, Spain). Clinical history, blood pressure (standard aneroid sphygmomanometer Model DS44, Welch Allyn, St. Louis, Mo, USA), and blood samples used to determine cholesterol, triacylglycerols, and glucose (SPIN 120, Barcelona, Spain) were also obtained using standard equipment. The mean blood pressure was calculated [$\left(\frac{Systolic-Diastolic}{3}\right)+Diastolic$] [18]. Using standardized procedures and equipment from reputable manufacturers, trained study staff collected anthropometric data following the Official Mexican Standard [19]. Height (Estadiometer SECA, model 213, Hamburg, Germany) and weight (digital scale Seca 700, Hamburg, Germany) measurements were used to determine body mass index (BMI; kg·m^−2^), and bioimpedance (Inbody 720, Bilbao, Spain) measurements were used to determine %BF.

Additionally, participants were given 15 min to acclimate to the cycle ergometer (Monark 828E, Vansbro, Sweden).

#### 2.3.2. Second Session

Before starting the maximal incremental exercise test on the cycle ergometer, the participants underwent 4 min of warm-up cycling at 0 Watts. Following the warm-up, the work rate was increased by 15 W·min^−1^ until the participants reached exhaustion while maintaining a pedaling rate of 50 rpm throughout the test [17]. To determine whether the exercise test was maximum, at least two of the following three fatigue indicators were considered: reaching the 9–10 level on the Borg Scale C-10 [20], inability to maintain 50 rpm for 10 s, and achieving ≥90% of their calculated maximum heart rate, which is 207-0.7 (age) [21]. Heart rate was continuously monitored during the tests using a POLAR monitor (model H7, Kempele, Finland) and VO_2_max was estimated using validated formulas: men: Y = 10.51 (Watts) + 6.35 (body mass, kg) − 10.49 (yr) + 519.3 mL·min^−1^; women: Y = 9.39 (Watts) + 7.7 (body mass, kg) − 5.88 (yr) + 136.7 mL·min^−1^ [22].

#### 2.3.3. Third and Fourth Sessions

The participants were randomly selected (random list in Excel Version 16.89.1) to complete two submaximal exercise tests using the cycle ergometer. Each test lasted 20 min and involved exercise at intensities of 60% and 80% of the maximum heart rate achieved during the maximum exercise test. The pedaling rate was maintained at 50 rpm throughout the test, and the workload was continuously adjusted to maintain a constant heart rate (±3 beats per minute).

### 2.4. Measurements of Concentrations of Cr and CysC

Blood samples were collected from the brachiocephalic vein while the participant was seated on the bike before and after each exercise test. The serum was then separated, divided into aliquots, and stored at −80 °C until further analysis. Serum Cr concentrations were determined using the Jaffé colorimetric kinetic method (creatinine kit from Spinreact, Sant Steve de Bas, Spain), and serum CysC levels were determined using the enzyme-linked immunosorbent assay (ELISA) technique (R&D system kit, Sant Steve de Bas, Spain). To account for the acute change in plasma volume due to physical exercise [23], post-exercise Cr and CysC values were adjusted for hemoconcentration [24]. Appendix A detailed the equations for estimating the GFR.

### 2.5. Statistical Analysis

The variances in errors and the normal distribution of the data were assessed using Levene’s and Shapiro–Wilk tests, respectively. The impact of the treatment (exercise tests) on blood concentrations of Cr, CysC, and eGFR was evaluated using the student’s *t*-test for related samples or the W-Wilcoxon test. The relationships between variables were examined using Pearson’s correlation. The effect size was calculated using Cohen’s d. The Bayes factor was used to quantify the marginal probability in favor of the null or alternative hypothesis. Statistical significance was set at *p* < 0.05. All analyses were conducted with Jamovi (Version 2.4, International License) and figures were created with Prism Version 9.5 (2022, GraphPad Software, Boston MA, USA).

## 3. Results

According to the analyzed health parameters (Table 1), the participants were in good health on average; however, they were overweight (26.8 ± 3.3 kg·m^−2^) and had low aerobic fitness (28.6 ± 7.1 mL·kg^−1^·min^−1^).

In addition, 7 of the 18 individuals had a resting eGFR of less than 60 mL·min^−1^·1.73 m^−2^ on at least two occasions (Appendix A).

The results show no evidence that physical exercise modified serum concentrations of CysC, Cr, or eGFR (Figure 3). On the contrary, the evidence for the null hypothesis was, in most cases, moderate: the treatment’s effect size (Cohen’s d) was <0.5 and the Bayes factor for the alternative hypothesis was <1 for all pairs of variables.

However, a negative correlation was observed between blood total cholesterol vs. eGFR by CysC (R = −0.512, R = −0.582, R = −0.531; *p* < 0.05) at rest, 60%, and 100% of the maximal heart rate measured during the maximal exercise test, respectively. Moreover, there was a negative correlation between age and eGFR by CysC at 60% of the intensity of the maximal heart rate (R = −0.516; *p* < 0.05).

## 4. Discussion

The central outcome of this study indicated that a single aerobic exercise on a bicycle and a short duration (≤20 min) at low (60%), moderate (80%), and high (100%) intensities did not affect serum concentrations of CysC, Cr, or eGFR in sedentary older adults (Figure 3). These results were evaluated using three statistics, p-value, effect size, and Bayes factor, confirming the lack of evidence for the alternative hypothesis (possible acute effect of exercise on CysC, Cr, or eGFR) [25,26]. However, increasing the sample size to confirm the results is highly recommended. Below are some physiological reasons for what was observed.

During physical exercise, the blood is redistributed throughout the different body compartments due to hemodynamic factors, mechanoreceptor stimulation caused by muscle contractions, and rapid sympathetic nervous system activation [27,28,29]. This compartment redistribution increases the blood flow to active muscles and decreases the flow to less active and visceral organs, including the kidneys [27]. Consequently, this redistribution could decrease the glomerular filtration rate (GFR), exacerbated at higher muscle contractions [30] or when exercise is performed in dehydrating climatic conditions [31], especially in older adults who have already experienced a decline in GFR [6]. Under these arguments, one might expect a decrease in GFR during exercise. However, it has been shown that the reduction in blood circulation at the glomerulus level and, consequently, in GFR only occurs for a few seconds at the onset of exercise [6], so this work did not observe modifications to the baseline values at the end of the exercise.

The stability of the eGFR across different exercise intensities observed in this study may be attributed to several physiological mechanisms. (1) The kidneys possess a robust autoregulatory system that helps maintain relatively constant renal blood flow and GFR despite fluctuations in systemic blood pressure during exercise [8]. (2) The 20 min exercise sessions in our study were not long enough to elicit significant changes in kidney function, unlike prolonged exercise sessions, which result in dehydration and reduced renal perfusion in older adults [7]. (3) Physical exercise at low intensities has protective effects on kidney function [3].

Our results align with those of Poussel et al. (2020), who found no significant GFR modifications after an extreme endurance event such as ultramarathons in highly trained individuals [15]. However, this differs from the results of Poortmans and Ouchinsky (2006), who reported a 30% reduction in GFR after a maximal exercise test in adults over 64 years [6]. This discrepancy could be attributed to (1) the exercise duration mentioned above; (2) participant characteristics: our sample was primarily comprised of healthy older adults, which may differ from other studies in terms of fitness levels or underlying health conditions; and (3) methodological differences: variations in the methods to estimate or measure GFR in different studies could result in discrepancies.

On the other hand, there is a natural and progressive decline in kidney function with age [32], primarily due to structural and vascular changes such as nephrosclerosis, loss of renal mass, and impaired angiogenesis [33]. These changes result in altered renal blood flow and a decreased GFR in this population [9]. The natural reduction in GFR is generally not clinically significant under normal conditions, and the kidney retains its functionality at an advanced age, but it could be critical during acute illnesses and pathological conditions. This study highlights no associations between age and eGFR at the basal level but there are associations at 60% of maximal exercise intensity. This tells us that GFR measurements during low-intensity exercise are more sensitive than those at rest.

According to the KDIGO guidelines [34], individuals with GFR <60 mL·min·1.73 m^2^ of body surface area experience chronic kidney disease; these values are linked to a high mortality ratio and fatal vascular events in individuals >70 years of age [35], besides affecting the medication doses and the management of several diseases [36]. In this sense, this study revealed a high prevalence of a low eGFR. Seven participants (38.9%) had a resting eGFR value <60 mL·min·1.73 m^2^ registered on two occasions. These findings align with the results of Abdulkader et al. (2017), who reported an impaired GFR in 19.3% of older adults, with 96.5% of those with associated comorbidities [37]. Additionally, Mohan et al. (2022) report that approximately 37 million adults in the United States are affected by chronic kidney disease [38]. Though a reduced resting GFR is considered normal and benign in aged individuals [8], regular monitoring of kidney function is necessary in these adults.

A secondary outcome of this study was the significant negative correlation observed at the end of the exercise tests between eGFR and total cholesterol. This demonstrated the high sensitivity of health status evaluation under stressed conditions. Several studies have observed an inverse correlation among eGFR, estimated by CysC, hypertriglyceridemia, and hypercholesterolemia [39,40]. The close relationship between CKD and increased cardiovascular disease risk (e.g., obesity, diabetes, hypertension, and aortic stenosis) has been proven elsewhere [35,37]. However, this work is one of the first to illustrate the connections in the context of acute exercise in older adults, which warrants further exploration.

Several mechanisms are described for the inverse relationship between low glomerular filtration rate and dyslipidemia. The main ones include decreased antioxidant and anti-inflammatory power with age and in people with chronic kidney disease, which increases oxidative stress, mitochondrial dysfunction, lipoprotein oxidation, and kidney damage [35,37,40].

The strengths of this study include controlled experiments, examining three exercise intensities, utilizing both Cr and CysC for GFR estimation, and focusing on older adults. Cr as a biomarker for kidney function is influenced by factors such as stress, muscle mass, and dietary protein intake, which may be particularly relevant in older adults [41]. Conversely, CysC is less affected by these factors and may provide a more accurate estimation of the GFR [42]. However, both biomarkers could provide a more comprehensive assessment of the effect of acute exercise on kidney function.

The limitations include the following: (1) A small sample size according to the initial program: the minimum sample size should be 44, with α < 0.05 (type I error) and β < 0.1 (type II error). (2) A short duration of submaximal exercise. (3) The absence of direct GFR measurement.

## 5. Conclusions

The results of this study indicate that short-duration exercise at low, moderate, and vigorous intensity does not significantly impact eGFR and is considered safe for kidney function in healthy older adults.

Given the inverse correlation between eGFR post-exercise and blood total cholesterol and the high prevalence of a low eGFR in older people, regular monitoring of kidney function in these people engaged in moderate- and high-intensity exercise is advised. This highlights the importance of tailoring exercise recommendations to individuals’ needs and specific abilities. However, a more comprehensive assessment with a larger sample and controlled studies is required.

## 6. Future Research Directions

It is recommended to increase the time (e.g., 60 min) or the exercise intensity (e.g., several cycles of moderate- and high-intensity intermittent exercise) because a single 20 min exercise session may not be sufficient to elicit a significant change in renal function concerning basal values in the studied population. Further investigation into the mechanisms that link blood cholesterol and eGFR during exercise is recommended. In addition, a focus should be placed on examining the impact of exercise on GFR in older adults with pre-existing kidney disease or cardiovascular conditions to identify the specific exercise-intensity levels at which kidney stress becomes more evident.

## Figures and Tables

**Figure 1 diseases-12-00249-f001:**
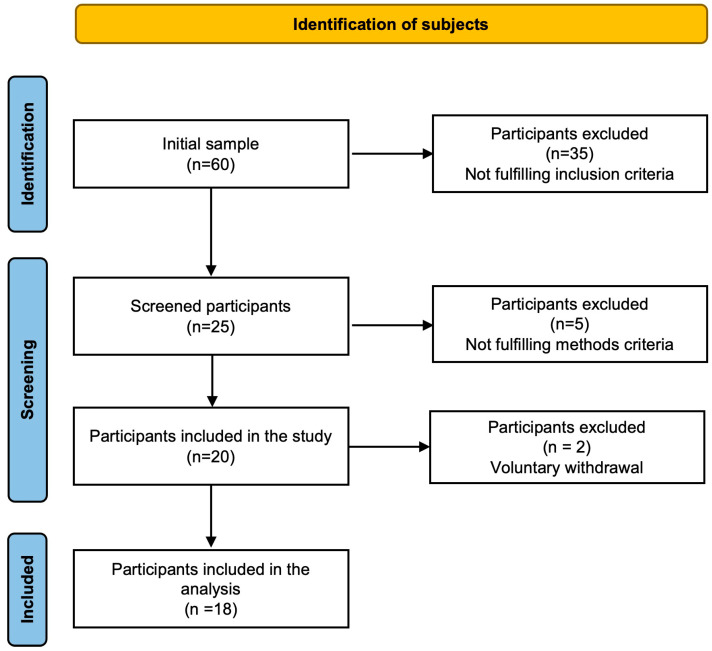
Screened participation.

**Figure 2 diseases-12-00249-f002:**
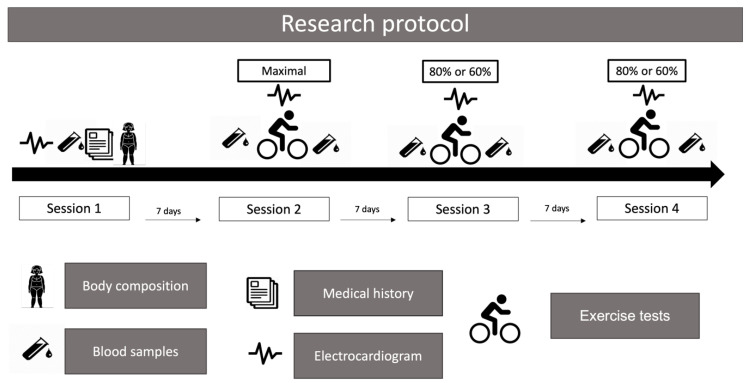
Research protocol.

**Figure 3 diseases-12-00249-f003:**
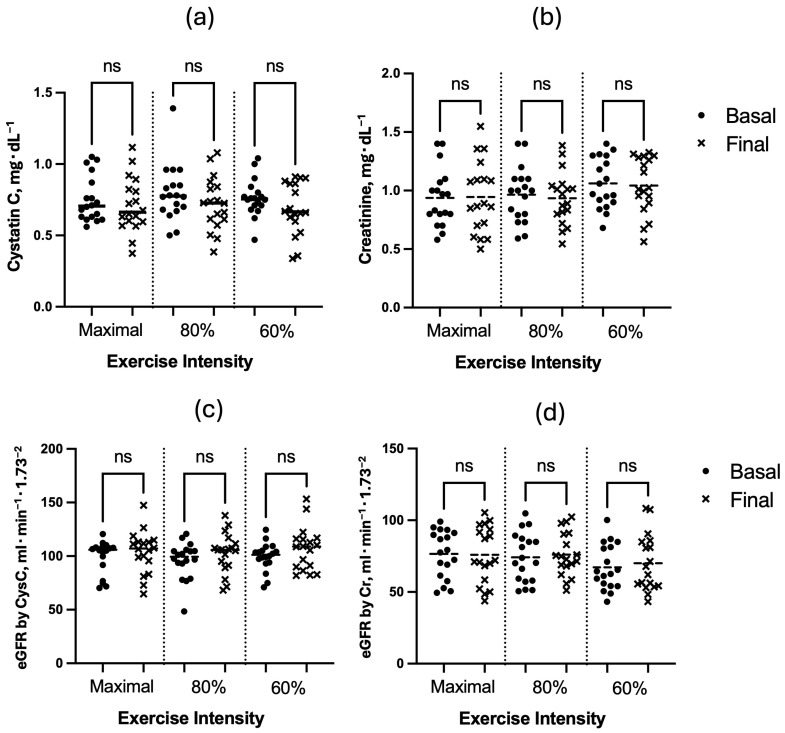
Effect of exercise tests on serum concentration of (**a**) Cystatin C, (**b**) creatinine, and estimated glomerular filtration rate (eGFR) calculated by serum concentration of (**c**) Cystatin C (CysC) and (**d**) creatinine (Cr).

**Table 1 diseases-12-00249-t001:** Participants’ characteristics.

Age (years)	70.1 ± 5.1
Weight (kg)	71.0 ± 10.9
Height (cm)	163.3 ± 7.6
BMI (kg·m^−2^)	26.8 ± 3.3
Fat mass (%)	24.1 ± 5.5
Systolic blood pressure (mmHg)	124.7 ± 13.1
Diastolic blood pressure (mmHg)	78.9 ± 10.2
Mean blood pressure (mmHg)	94.2 ± 10.3
Glucose (mg·dL^−1^)	103 ± 13
Cholesterol (mg·dL^−1^)	185 ± 34
Triacylglycerols (mg·dL^−1^)	127 ± 41
Creatinine (mg·dL^−1^)	0.937 ± 0.246
Cystatin C (mg·dL^−1^)	0.753 ± 0.161
eGFR by Cr (mL·min·1.73 m^2^)	76.5 ± 16.7
eGFR by CysC (mL·min·1.73 m^2^)	99.3 ± 15.4
VO_2_ max (ml·kg^−1^·min^−1^)	28.6 ± 7.1
HRmax (beats·min^−1^)	148 ± 13

BMI: body mass index, Cr = creatinine, CysC = Cystatin C, eGFR = estimated glomerular filtration rate, HRmax: maximum heart rate, VO_2_ max: maximum oxygen consumption. Values are mean ± SD.

## Data Availability

Everyone who contacts the corresponding author will have access to the required data (aramos@uacj.mx).

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
