# Peer review of "Acute Effect of Three Aerobic Exercise Intensities on Glomerular Filtration Rate in Healthy Older Adults"

_diseases, 2024, doi:10.3390/diseases12100249_

Round 1
Reviewer 1 Report
Comments and Suggestions for Authors
In this manuscript, the authors summarized and discussed the acute effect of three aerobic exercise intensities on the glomerular filtration rate in healthy, sedentary older adults. The manuscript is well-organized and clearly stated. I would suggest accepting it after the following major concern is addressed.
1. In line 62, “18 participants (12 men and eight women)” were claimed as participants. Please check the number again and write in a unified format (all in Arabic numerals or English).
2. As for healthy adults, it is clear that kidneys restore certain abilities for compensation when they get injured before there’s a change at CysC, Cr, or eGFR. Therefore, it could be predicted that negative results would probably be found in this research, especially for short-time aerobic exercises. In such a context, investigations in older adults who have already experienced a decline in eGFR would be more meaningful. Why did the authors choose to investigate such a project and please state the significance and potential of this study.
3. A single 20-minute exercise session may not be long enough to elicit significant change in kidneys. Prolonged exercise time or multiple exercise sessions would be more suitable for investigation.
Author Response
In this manuscript, the authors summarized and discussed the acute effect of three aerobic exercise intensities on the glomerular filtration rate in healthy, sedentary older adults. The manuscript is well-organized and clearly stated. I would suggest accepting it after the following major concern is addressed.
R: Thank you
In line 62, “18 participants (12 men and eight women)” were claimed as participants. Please check the number again and write in a unified format (all in Arabic numerals or English).
R: Thank you. The line was corrected.
As for healthy adults, it is clear that kidneys restore certain abilities for compensation when they get injured before there’s a change at CysC, Cr, or eGFR. Therefore, it could be predicted that negative results would probably be found in this research, especially for short-time aerobic exercises. In such a context, investigations in older adults who have already experienced a decline in eGFR would be more meaningful. Why did the authors choose to investigate such a project and please state the significance and potential of this study.
R: Thank you. As you suggested, valuable information was added throughout the introduction, highlighting the study's purpose and importance.
A single 20-minute exercise session may not be long enough to elicit significant change in kidneys. Prolonged exercise time or multiple exercise sessions would be more suitable for investigation.
R: Thank you. Your suggestion was added in section 6: Future research directions.
Reviewer 2 Report
Comments and Suggestions for Authors
The manuscript entitled “Acute effect of three aerobic exercise intensities on glomerular filtration rate in healthy older adults” was aimed to assess the immediate effects of three aerobic exercise intensities on eGFR in healthy, sedentary older adults using serum concentrations of Cr and CysC. The topic is relevant and the authors obtained important data that should be considered at aerobic exercising by old people. Thus, this determines high potential interest of readers. The experimental part is well-designed and the manuscript deserves consideration after minor revision. The main comments and recommendations are listed below.
– Details for equipment and software should be unified (Manufacturer, City, Country). Models can be given out of brackets
– Provide discussion on all parameters given in Table 1
– Support conclusions by the corresponding data obtained
– L. 271-272. Mention all supplementary items
Author Response
The manuscript entitled “Acute effect of three aerobic exercise intensities on glomerular filtration rate in healthy older adults” was aimed to assess the immediate effects of three aerobic exercise intensities on eGFR in healthy, sedentary older adults using serum concentrations of Cr and CysC. The topic is relevant and the authors obtained important data that should be considered at aerobic exercising by old people. Thus, this determines high potential interest of readers. The experimental part is well-designed and the manuscript deserves consideration after minor revision. The main comments and recommendations are listed below.
– Details for equipment and software should be unified (Manufacturer, City, Country). Models can be given out of brackets
R: Thank you. The requested information has been added.
– Provide discussion on all parameters given in Table 1
R: Thank you. Information on possible mechanisms for the relationship between low glomerular filtration rate and dyslipidemia was added.
– Support conclusions by the corresponding data obtained.
R: Thank you. Your recommendation was considered.
– L. 271-272. Mention all supplementary items
R: Thank you. This section is managed by the journal's editorial committee.
Reviewer 3 Report
Comments and Suggestions for Authors
The study design is described and clearly explained in the introduction. The sample size (n=18), however, is very small and casts more than one doubt to the sustainability of the study results. We suggest to increase, if possible, the sample size or to include a discussion on the statistical meaning of results; even if a statistical analysis paragraph is already included in Methods.
Apart from a few explanations in Discussion, it is unclear how cholesterol, triglycerides and glucose concentrations were useful or important in assessing patients status in relation to the eGFR evaluation. Additional biochemical parameters could have provided better insights on the renal function status of patients, on their hemodynamic function, etc.. We suggest, whether still possible, to provide additional parameters in patients basal evaluation of, at least, to better discuss why those parameters were chosen.
A sentence such as “The correlations detected between eGFR post-exercise
and blood cholesterol may require a more comprehensive assessment and controlled renal
function studies post-acute exercise.” is probably not suited for Conclusions and should be evaluated in discussion.
The sentence “the importance of tailoring exercise recommendations to individuals' needs and specific abilities” is very general and not contextualized in the specific study framework. We suggest also addressing the importance of tailoring exercise to study participants in the Introduction, not in conclusions.
It is unclear how “This work will contribute to determining the most effective exercise regimen for preserving or enhancing kidney health in older individuals” seems not supported by the study results whereas the goal of “provide reassurance regarding exercise recommendations for this demographic group” appears backed-up by the study results, despite the small numner of subjects examined.
Author Response
The study design is described and clearly explained in the introduction. The sample size (n=18), however, is very small and casts more than one doubt to the sustainability of the study results. We suggest to increase, if possible, the sample size or to include a discussion on the statistical meaning of results; even if a statistical analysis paragraph is already included in Methods.
R: Thank you. Following your suggestion, new analyses were carried out, and the implications of the sample were discussed:
- In the methods section, statistics are added that are recommended for a small sample.
- In the results section, the values of the statistics are presented.
- In the discussion section, the evidence that supports the results is presented.
Apart from a few explanations in Discussion, it is unclear how cholesterol, triglycerides and glucose concentrations were useful or important in assessing patients status in relation to the eGFR evaluation. Additional biochemical parameters could have provided better insights on the renal function status of patients, on their hemodynamic function, etc.. We suggest, whether still possible, to provide additional parameters in patients basal evaluation of, at least, to better discuss why those parameters were chosen.
R: Thank you. The methods section clarifies the reason for the clinical analyses performed on the participants. In addition, the discussions explain the importance of the associations found.
A sentence such as “The correlations detected between eGFR post-exercise and blood cholesterol may require a more comprehensive assessment and controlled renal function studies post-acute exercise.” is probably not suited for Conclusions and should be evaluated in discussion.
R: Thank you. Your suggestion was considered.
The sentence “the importance of tailoring exercise recommendations to individuals' needs and specific abilities” is very general and not contextualized in the specific study framework. We suggest also addressing the importance of tailoring exercise to study participants in the Introduction, not in conclusions.
R: Thank you. Your suggestion was considered.
It is unclear how “This work will contribute to determining the most effective exercise regimen for preserving or enhancing kidney health in older individuals” seems not supported by the study results whereas the goal of “provide reassurance regarding exercise recommendations for this demographic group” appears backed-up by the study results, despite the small numner of subjects examined.
R: Thank you. Your suggestion was considered.
Round 2
Reviewer 1 Report
Comments and Suggestions for Authors
In this manuscript, the authors summarized and discussed the acute effect of three aerobic exercise intensities on the glomerular filtration rate in healthy, sedentary older adults. The manuscript is well-organized and clearly stated. The major concerns were addressed and I would suggest accepting it.
Author Response
Thank you for your comments and recommendation
Reviewer 3 Report
Comments and Suggestions for Authors
The paper has been considerably improved and our previous criticisms have been addressed. However, we think that the abstract should be updated to reflect current manuscript updates.
Author Response
The paper has been considerably improved and our previous criticisms have been addressed. However, we think that the abstract should be updated to reflect current manuscript updates.
R: Thank you. Your recommendation was considered.